# Implementation of an Enhanced Recovery after Surgery Protocol in Advanced and Recurrent Rectal Cancer Patients after beyond Total Mesorectal Excision Surgery: A Feasibility Study

**DOI:** 10.3390/cancers15184523

**Published:** 2023-09-12

**Authors:** Stefi Nordkamp, Davy M. J. Creemers, Sofie Glazemakers, Stijn H. J. Ketelaers, Harm J. Scholten, Silvie van de Calseijde, Grard A. P. Nieuwenhuijzen, Jip L. Tolenaar, Hendi W. Crezee, Harm J. T. Rutten, Jacobus W. A. Burger, Johanne G. Bloemen

**Affiliations:** 1Department of Surgery, Catharina Hospital, 5623 EJ Eindhoven, The Netherlandsgrard.nieuwenhuijzen@catharinaziekenhuis.nl (G.A.P.N.);; 2Department of GROW, School for Oncology and Reproduction, Maastricht University, 6229 ER Maastricht, The Netherlands; 3Department of Anaesthesiology, Catharina Hospital, 5623 EJ Eindhoven, The Netherlands

**Keywords:** Enhanced Recovery After Surgery, rectal cancer, locally advanced rectal cancer, locally recurrent rectal cancer, surgery

## Abstract

**Simple Summary:**

The aim of this study was to evaluate the implementation and outcomes of a tailored ERAS LARRC protocol developed for patients with locally advanced and recurrent rectal cancer who require complex surgical procedures. This study shows that the protocol is feasible with a compliance rate of 73.6% and results in a reduction in postoperative complications. Multimodal anaesthesia could potentially impact the length of stay in a beneficial way, as well as improve the recovery profile.

**Abstract:**

Introduction: The implementation of an Enhanced Recovery After Surgery (ERAS) protocol in patients with locally advanced rectal cancer (LARC) and locally recurrent rectal cancer (LRRC) has been deemed unfeasible until now because of the heterogeneity of this disease and low caseloads. Since evidence and experience with ERAS principles in colorectal cancer care are increasing, a modified ERAS protocol for this specific group has been developed. The aim of this study is to evaluate the implementation of a tailored ERAS protocol for patients with LARC or LRRC, requiring beyond total mesorectal excision (bTME) surgery. Methods: Patients who underwent a bTME for LARC or LRRC between October 2021 and December 2022 were prospectively studied. All patients were treated in accordance with the ERAS LARRC protocol, which consisted of 39 ERAS care elements specifically developed for patients with LARC and LRRC. One of the most important adaptations of this protocol was the anaesthesia procedure, which involved the use of total intravenous anaesthesia with intravenous (iv) lidocaine, iv methadone, and iv ketamine instead of epidural anaesthesia. The outcomes showed compliance with ERAS care elements, complications, length of stay, and functional recovery. A follow-up was performed at 30 and 90 days post-surgery. Results: Seventy-two patients were selected, all of whom underwent bTME for either LARC (54.2%) or LRRC (45.8%). Total compliance with the adjusted ERAS protocol was 73.6%. Major complications were present in 12 patients (16.7%), and the median length of hospital stay was 9 days (IQR 6.0–14.0). Patients who received multimodal anaesthesia (75.0%) stayed in the hospital for a median of 7.0 days (IQR 6.8–15.5). These patients received fewer opioids on the first three postoperative days than patients who received epidural analgesia (*p* < 0.001). Conclusions: The implementation of the ERAS LARRC protocol seemed successful according to its compliance rate of >70%. Its complication rate was substantially reduced in comparison with the literature. Multimodal anaesthesia is feasible in beyond TME surgery with promising effects on recovery after surgery.

## 1. Introduction

The colorectal Enhanced Recovery After Surgery (ERAS) protocol is currently used as the standard mode of care in patients requiring surgical treatment for colorectal cancer [1]. To improve surgical outcomes, a compliance rate of >70% to ERAS care elements has been associated with improved outcomes in minimally invasive colorectal cancer surgery [2]. In patients with locally advanced rectal cancer (LARC) and locally recurrent rectal cancer (LRRC), the implementation of the ERAS protocol has been deemed challenging due to the complexity and heterogeneity of this disease and treatment [3].

Patients with LARC or LRRC require neoadjuvant chemotherapy and radiotherapy, followed by a beyond total mesorectal excision (bTME). This procedure consists of a TME procedure, with an extended resection of the sacral and/or lateral pelvic wall and often multivisceral organ resections [4,5]. As a result, prolonged lengths of hospital stays and higher rates of morbidity and mortality (Clavien–Dindo ≥ III of 20–40%) are described in these patients [6,7,8,9]. In an earlier study, it was observed that patients with LARC and LRRC are a substantially different group regarding ERAS compliance and postoperative outcomes compared to patients with non-advanced colorectal cancer [10].

Due to the increasing level of experience with ERAS principles in colorectal surgery, it seemed the appropriate time to apply the ERAS principles to advanced rectal cancer surgery. In several high-expertise fields, such as upper gastro-intestinal surgery and cytoreductive surgery with hyperthermal intraperitoneal chemotherapy, ERAS implementation has seemed promising, with good postoperative results [7,11,12,13]. Therefore, a tailored ERAS protocol for locally advanced and recurrent rectal cancer (LARRC) has been developed by a multidisciplinary team with expertise in advanced rectal cancer [10]. The LARRC protocol was based on the colorectal and pelvic ERAS protocol of Gustafsson [1] and Nygren [12], as well as the pelvic exenteration protocol of Harji et al. [7]. Specific adaptations were made in the anaesthesia protocol, oral intake, postoperative mobilisation, and urological care pathways, as well as strict guidelines for the use of drains and catheters to suit the needs of these patients.

The aim of this study was to evaluate the implementation and outcomes of the tailored ERAS LARRC protocol developed for patients with locally advanced and recurrent rectal cancer who require bTME.

## 2. Materials and Methods

All consecutive patients with LARC or LRRC who underwent a bTME with curative intent in the Catharina Hospital Eindhoven, a tertiary referral hospital for rectal cancer, were included in this study from October 2021 to December 2022. After the development of the ERAS LARRC protocol in July 2021, a period of 3 months was used for the initiation phase; all involved caretakers (surgeons, anaesthesiologists, intensivists, intensive care nurses, ward nurses, nurse practitioners, stoma care nurses, dieticians, physiotherapists, and surgical residents) were educated to standardise the (digital) system for the care of these patients in the new protocol.

### 2.1. Patients and Treatment

LARCs were standardly treated with neoadjuvant (chemo)radiotherapy, whereas LRRC patients underwent neoadjuvant chemoradiotherapy or chemo re-irradiation in the case of previous pelvic irradiation [14]. Some patients received induction chemotherapy before chemoradiotherapy [15,16]. After neoadjuvant treatment, a bTME was performed. In this study, a bTME was defined as a total mesorectal excision, with the resection of the sacral and/or lateral pelvic wall and/or multivisceral resections, including partial or total pelvic exenterations. Surgery was often combined with intraoperative radiotherapy (10–12.5 Gy) for the margins considered at risk. Other surgical specialists were consulted if urological, plastic, or vascular reconstructions were required.

### 2.2. Enhanced Recovery after Surgery Protocol

All patients were treated in accordance with the ERAS LARRC protocol, as shown in Appendix A The ERAS LARRC protocol. It consisted of 39 newly developed ERAS care elements that were analysed for their calculation of compliance and based on colorectal care elements in the EIAS© system [10]. One of the most important adaptations of this protocol was the implementation of a different anaesthesia procedure, which involved the use of total intravenous anaesthesia with intravenous (iv) lidocaine, iv methadone, and iv ketamine, instead of epidural anaesthesia [17,18,19]. During surgery, continuous wound infusion catheters were applied, which stayed in place for the first two to three postoperative days [20]. As patients commonly suffer from urinary retention after extensive rectal surgery, the use of suprapubic catheters was encouraged. As many of these patients receive urological reconstructions, and postoperative paralytic ileus seems to be associated with radical cystectomies, these patients received postoperative gastric tubes, while all others did not [21,22,23]. A strict removal procedure followed, and all details about production and removal were registered for analysis. Every patient had a bedside map attached to their bed, along with the information on the protocol (Figure 1).

### 2.3. Data Collection and Follow-Up

Patient and tumour characteristics, data on ERAS care elements, complications, and functional recovery (e.g., time until first passage of stool, mobilisation, and length of hospital stay) were prospectively collected from the medical records. Preadmission ERAS elements (e.g., patient education, optimisation of patient’s health status), preoperative ERAS elements (e.g., antibiotic and perioperative nausea and vomiting prophylaxis), intraoperative ERAS elements (e.g., anaesthesia, blood loss, duration of procedure and fluid management), and postoperative ERAS elements (e.g., nasogastric tube management, drain management, pain management, and oral intake) were collected. Complications occurring during the first 30 and 90 postoperative days were scored using the Clavien–Dindo classification [24].

### 2.4. Statistical Analyses

Statistical analyses were performed using SPSS Statistics 29.0 software (IBM, Endicott, NY, USA). The primary endpoints were the percentage of ERAS compliance in comparison to the ERAS LARRC protocol, time to functional recovery, and postoperative complications. Complications were classified using the Clavien–Dindo classification and divided into minor complications (Clavien–Dindo I–IIIa) and major complications (Clavien–Dindo IIIb–IV). Compliance with the protocol was calculated based on the 39 developed ERAS care elements. Secondary endpoints included ERAS-related outcomes per perioperative phase. Demographics were presented for all patients. Sub-analyses were included for the performed anaesthesia and the use of a nasogastric tube because of the diversity of hypotheses in the literature regarding these elements in ERAS care. Continuous data were reported as means with standard deviations or as medians with ranges, depending on parameter distribution. Categorical data were reported as the count with percentages. Group comparisons were performed using the Chi-square test, Fisher’s exact test, or the Mann–Whitney U test, as appropriate.

## 3. Results

A total of 72 patients with rectal cancer underwent bTME surgery between October 2021 and December 2022, of whom 39 (54.2%) were LARC patients and 33 (45.8%) were LRRC patients. Induction chemotherapy was administered in 36 patients (50.0%), while all patients received neoadjuvant radiotherapy. In total, 48 patients (66.7%) received full-course chemoradiotherapy with 50–50.4 Gy, 20 patients (27.8%) received chemoreirradiation with 30–30.6 Gy, and 4 patients (5.5%) received short-course radiotherapy with 25 Gy. Of the bTME surgical procedures, 23 patients (31.9%) underwent pelvic exenteration, 27 patients (37.5%) underwent abdomino-perineal resection (APR), and 18 patients (25.0%) underwent resection with primary (re-)anastomosis. In total, 14 patients (19.4%) underwent additional sacral resection, and 36 patients (50.0%) underwent additional pelvic sidewall resection. The median time of surgery was 306.0 min with a median of 1550.0 mL blood loss. All patient and treatment characteristics are shown in Table 1.

### 3.1. Compliance to the ERAS LARRC Protocol

Total compliance with the ERAS LARRC protocol was 73.6%, in which all 39 items were scored per ERAS care element, as shown in Table 2. The preadmission compliance was 81.9%, the preoperative compliance was 89.4%, the intraoperative compliance was 70.2%, and the postoperative compliance was 62.8% for the total group.

### 3.2. Comparison of ERAS-Related Outcomes: Anaesthesia and Pain Management

Fifty-four patients (75.0%) received multimodal anaesthesia and postoperative continuous wound infusion (CWI) catheters, while, in 18 patients (25.0%), epidural anaesthesia was used. The use of opioids during the first three postoperative days was significantly lower in the multimodal anaesthesia group (*p* < 0.001) when converted to morphine equivalents. Patients had comparable scores on the numeric pain rating scale (NRS) during the first three postoperative days.

Of the patients with epidural anaesthesia, 17 patients (94.4%) stayed in the ICU compared to 28 patients (51.9%) with multimodal anaesthesia (*p* = 0.001). Of the patients receiving epidural anaesthesia, 14 patients (82.4%) needed hemodynamic support compared to 10 patients (35.7%) in the multimodal anaesthesia group (*p* = 0.002).

Patients with an epidural stayed in the hospital for a median of 10 days (IQR 6.8–15.5) compared to 7.0 days (5.5–14.0) for patients treated with multimodality analgesia (*p* = 0.440) (Table 3).

### 3.3. Comparison of ERAS-Related Outcomes: Nasogastric Tube Management

A nasogastric tube was placed in 39 patients (54.2%). Among patients requiring a nasogastric tube for a postoperative period longer than 2 days, 13 patients (50.0%) underwent a total pelvic exenteration and a urologic reconstruction, and 23 patients (88.5%) underwent omentoplasty. The median production from the nasogastric tube during the first three postoperative days was 15.0 mL (IQR 0.0–80.0), 50.0 mL (0.0–260.0), 290.0 mL (50.0–800.0), and 300.0 mL (85.0–2025.0), respectively. A nasogastric tube was placed for a median of 3.0 days (IQR 2.0–6.5).

### 3.4. Comparison of ERAS-Related Outcomes: Urological Management

In 38 patients (52.8%), urethral catheters were placed intraoperatively and used after surgery. These urethral catheters were in place for a median of 6.0 days (IQR 3.0–12.0). In five patients (6.9%), the catheter was replaced due to bladder retention. A total of 10 patients (13.9%) had a urethral catheter at discharge, mostly following psoas hitch reconstruction (*p* = 0.040). A total of 8 patients (11.1%) received a suprapubic catheter, and 4 of these patients were discharged with the catheter. Twenty-three patients (31.9%) had an urostomy because of a total pelvic exenteration.

### 3.5. Outcomes, Functional Recovery, and Complications

The median hospital stay was 9 days (IQR 6.0–14.0). On the day of surgery, 42 patients (58.3%) were mobilised according to the protocol (out of bed for 5–15 min). On the third postoperative day, 56 patients (77.8%) were mobilised according to the protocol (they were out of bed twice a day for a minimum of one hour and walked around the ward (4–5 times 100 m)).

In total, 51 patients (70.8%) suffered from postoperative complications. Within 30 days after surgery, major complications (Clavien–Dindo > IIIa) were observed in 12 patients (16.7%) and between 31 and 90 days after surgery in 7 patients (9.7%). Most complications were of gastrointestinal origin. Of the total group of patients with complications, 14 patients (19.4%) needed surgical intervention, and 19 patients (26.4%) were readmitted to hospital (Table 4).

## 4. Discussion

This study demonstrates the feasibility of implementing an ERAS protocol specifically designed for patients undergoing bTME for LARC or LRRC with a compliance rate of 73.6%. Patients treated within the ERAS LARRC protocol had lower rates of postoperative complications in comparison to the literature [6,7,8,9]. The use of multimodal analgesia, instead of epidural analgesia, was effective in postoperative pain management, with potentially beneficial effects on functional recovery and length of stay.

In the literature, evidence for the beneficial effects of an ERAS protocol for patients with LARC or LRRC undergoing bTME is lacking, as this heterogeneous and complex patient group is commonly excluded from ERAS-related studies [25]. The potential benefit of an ERAS protocol in these patients could be extensive as they suffer from poor postoperative outcomes due to a long period of functional recovery and a high complication rate. However, to expect an improvement in postoperative outcomes and time to functional recovery via ERAS implementation, compliance rates of at least 70% appear necessary [26,27].

The development and implementation of a new ERAS protocol is challenging. The implementation of the ERAS LARRC protocol in this centre was facilitated due to the prior implementation of an ERAS protocol for non-advanced colorectal surgical care. Certain elements could be implemented directly, such as proactive education and nursing pathways, careful fluid management, and attention to nausea and vomiting, as these elements have a significant effect on functional recovery after any gastrointestinal surgical procedure [5]. Other elements could not be implemented directly or had to be altered, taking into account the extensive surgical procedure and the increased likelihood of postoperative morbidity in LARRC patients. In the minimally invasive colorectal ERAS protocol, epidural analgesia is obsolete, but in beyond TME surgery for LARC or LRCC, it is considered necessary for adequate postoperative pain management [7]. In our multidisciplinary team, a novel multimodal approach with the use of methadone, lidocaine, and ketamine was proposed instead of using epidural analgesia. The use of multimodal analgesia was implemented gradually during the study period with promising results.

Epidural analgesia is not easily changed to a multimodal approach, as it remains the golden standard in treating postoperative pain. At the beginning of the studied period, all patients with sacral resections received epidural analgesia, while during implementation, more patients received multimodal analgesia with increasingly promising results. This resulted in more affinity with the multimodal approach and a reduction in used epidural analgesia. However, selection bias was not prevented and should be investigated in future research.

Even so, the implementation of the adjusted ERAS protocol seems feasible, and the postoperative results are also encouraging. Comparing the outcomes of this study to other examples in the literature in terms of ERAS care elements is challenging, and, to our knowledge, ERAS has not yet been implemented in patients with LARC or LRRC. One ERAS implementation study conducted by Harji et al. investigated the implementation of an ERAS protocol adjusted specifically for patients with pelvic malignancies undergoing pelvic exenteration [7]. A few other studies have presented postoperative outcomes after bTME in rectal cancer, investigating outcomes after pelvic exenteration for pelvic malignancies of any kind without the implementation of a specific ERAS protocol [6,28,29,30,31,32]. In these studies, patient characteristics and procedures were comparable to this study, such as operating time and blood loss. The length of stay in these studies varied between 9 and 19 days, while in this study, the median hospital stay was 9 days. Stays in the intensive care unit were reduced, as patients remained there for 1 to 2 days, compared to 3 to 4 days in other studies. In the literature, the major complication rate (Clavien–Dindo > IIIa) showed a median of 22.6% to 61.3%, compared to a median major complication rate of 16.7% in this cohort. In conclusion, the results of this study seem rather promising.

Based on the results of this study, the presented ERAS LARRC protocol is implementable and valuable and can be applied in clinical practise. However, the current protocol still should be tailored to some care elements. The heterogeneity of this disease and type of surgical procedure complicates implementation, and some exceptions or deviations from protocol are inevitable. The quality of the protocol lies within the combination of all individual ERAS care elements, and successful implementation requires continuous effort, feedback, and further development. As with all new protocol implementations, compliance increases over time by gaining more affinity and experience with the protocol. In our centre, caretakers were already familiar with ERAS, which facilitated the implementation. Even so, as compliance was already >70%, it should increase even further over time due to the increased affinity of caretakers with it, as well as the continuous evolution of the protocol.

It is essential to identify factors that are associated with compliance and functional recovery in patients with LARC and LRRC. Postoperative care elements have the greatest impact on overall recovery, yet they are also the most difficult to comply with [7]. This was reflected in this study. Even though the overall compliance rate was >70%, postoperative compliance was only 62.8%. This was mainly due to an inability to attain a sufficient amount of oral intake and the inability to weigh patients postoperatively in order to manage the fluid balance. As 54.2% of patients had a nasogastric tube postoperatively, the appropriate intake could not be accomplished in these patients. As most patients seemed to develop gastroparesis and/or paralytic ileus on the second postoperative day, it appears possible to abstain from using a nasogastric tube for the first two postoperative days in most patients. A potential additional benefit of omitting a nasogastric tube may be that patients are less constrained in postoperative mobilisation. As oral intake is increased gradually within the ERAS LARRC protocol, a nasogastric tube could still be inserted in case of gastroparesis and/or paralytic ileus. Another future adjustment is the promotion of suprapubic catheters as an alternative to urethral catheters in case of expected bladder retention after surgery. However, a patient cohort that might benefit from a suprapubic catheter cannot be defined based on the current study. In upcoming years, implementation in other centres is needed to investigate all specific care elements and guarantee an effective ERAS LARRC protocol as the new standard of care.

## 5. Conclusions

This study shows that an ERAS protocol developed for patients undergoing bTME for LARC or LRRC is feasible. In comparison to prior studies in the literature, the current ERAS LARRC protocol results in a reduction in complications. Multimodal anaesthesia could potentially impact the length of stay in a beneficial way, as well as improve the recovery profile. Despite the fact that the presented cohort study is a work in progress, it shows that even in this heterogenic group, standardisation in perioperative care is achievable and may yield promising results.

## Figures and Tables

**Figure 1 cancers-15-04523-f001:**
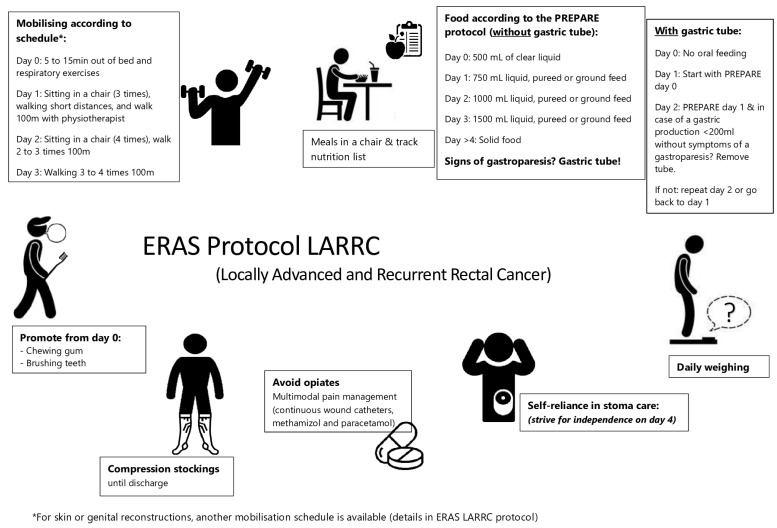
Bed side map of the ERAS LARRC protocol.

**Table 1 cancers-15-04523-t001:** Baseline characteristics.

Baseline Characteristics	*n* = 72 (%)
Age, mean in years (SD)	63.9 (10.7)
Gender	
Male	48 (66.7)
Female	24 (33.3)
ASA class	
I–II	59 (82.0)
III–IV	13 (18.1)
Induction chemotherapy	36 (50.0)
Neoadjuvant radiotherapy	72 (100)
Pathological Tumour stage	
T0–T2	6 (8.3)
T3–T4	37 (51.4)
N.A. *	29 (40.3)
Pathological Nodal stage	
N0	26 (36.1)
N1/2	17 (13.4)
N.A. *	29 (40.3)
Main procedure	
LAR/Re-resection with anastomosis	18 (25.0)
APR	27 (37.5)
Total exenteration	23 (31.9)
Tumour resection n.o.s. **	23 (31.9)
Intraoperative radiotherapy	58 (80.6)
Bladder resection	24 (33.3)
Urologic reconstruction	34 (47.2)
Partial sacral resection	14 (19.4)
Prostate and/or vesicles resection (male only)	35 (72.9)
Uterus and/or ovaria resection (female only)	13 (54.2)
Vagina resection (female only)	7 (29.2)
Pelvic sidewall resection	36 (50.0)
Lateral lymph node resection	9 (12.5)
Length of operation, median in mL (IQR)	306.0 (219.3–368.8)
Intraoperative blood loss, median in mL (IQR)	1550.0 (762.5–2875.0)
Omentoplasty	51 (70.8)

* N.A. = not applicable in patients with LRRC. ** n.o.s. = not other specified.

**Table 2 cancers-15-04523-t002:** Compliance to ERAS LARRC protocol.

	Preadmission Compliance Care Elements	Compliance %
1	Preoperative nutritional status assessed	98.6
2	Preoperative nutritional treatment in case of (risk for) malnutrition	14.3
3	Alcohol (quitted before surgery)	93.1
4	Preadmission patient education	94.4
5	Ostomy introduction	100
6	Patient screened for anaemia preoperatively	94.4
7	Anaemia treatment given when applicable	60
8	Smoker (quitted before surgery)	81.9
9	Prehabilitation (in case of vulnerability)	100
	**Preoperative compliance care elements**	
10	No oral bowel preparation used unless patients received an LAR	94.4
11	Preoperative oral carbohydrate treatment	91.7
12	No preoperative sedative medication < 65 years	95.8
13	Thrombosis prophylaxis administered until outpatient at 28 days + compression socks	77.8
14	Antibiotic prophylaxis before incision	100
15	PONV prophylaxis administered	100
16	Date of admission = date of surgery (unless they received ureter stents)	81.9
17	SDD administered	73.6
	**Intraoperative compliance care elements**	
18	No epidural or spinal anaesthesia but use of multimodal anaesthesia	75
19	Nerve blocks or local anaesthesia and continuous wound infusion	75
20	Forced-air heating cover used	100
21	No nasogastric tube placed intraoperatively and used postoperatively unless ileus appeared; it was then removed according to protocol	44.2
22	No resection-site drainage placed according to protocol (pelvic exenteration)	56.9
	**Postoperative compliance care elements**	
23	Termination of urinary drainage at end of operation; SPC was used in case of potential retention bladder, Otherwise it was handled according to protocol	84.7
24	Stimulation of gut motility: laxatives and non-medicamental treatment used according to protocol	100
25	Weighted on POD 0	43.1
26	Weighted on POD 1	29.2
27	Weighted on POD 2	65.3
28	Weighted on POD 3	72.2
29	Pain management with CWI, metamizole and paracetamol	72.2
30	PONV prevention	100
31	Energy intake on POD 0 > 500 ml	20.8
32	Energy intake on POD 1 > 750 ml	43.1
33	Energy intake on POD 2 > 1000 mL	37.5
34	Energy intake on POD 3 > 1500 mL	41.7
35	Mobilisation on day of surgery	58.3
36	Mobilisation on POD1 according to protocol	52.8
37	Mobilisation on POD2 according to protocol	69.4
38	Mobilisation on POD3 according to protocol	77.8
39	Follow-up control performed around 30 days postoperatively	100

POD = Postoperative day.

**Table 3 cancers-15-04523-t003:** Anaesthesia protocol and postoperative pain management.

Anaesthesia	Multimodal Management	Epidural	
	*n* = 54 (75.0%)	*n* = 18 (25.0%)	*p*-Value
Postoperative analgesics			
Continuous wound infusion	53 (98.1)	1 (5.6)	<0.001
Patient-controlled analgesia (PCA)	16 (29.6)	8 (44.4)	0.456
Median duration of PCA in days (IQR)	4.0 (3.0–5.0)	3.0 (1.0–4.0)	0.269
Epidural	NA	3.0 (2.0–3.0)	
Postoperative NSAID (Metamizole iv)	49 (90.7)	13 (72.2)	0.063
Median duration of metamizole in days (IQR)	5.0 (3.0–6.0)	5.0 (3.0–6.5)	0.641
Postoperative opioids * POD 0	53 (98.1)	18 (100)	0.750
Postoperative opioids * POD 1	23 (42.6)	18 (100)	<0.001
Postoperative opioids * POD 2	15 (27.8)	18 (100)	<0.001
Postoperative opioids * POD 3	14 (25.9)	17 (94.4)	<0.001
Postoperative opioids * POD 4	13 (24.1)	11 (64.7)	0.002
Postoperative opioids * POD 5	12 (22.2)	7 (41.2)	0.112
Nausea POD 0	10 (18.6)	1 (5.6)	0.380
Nausea POD 1	15 (27.8)	3 (16.7)	0.314
Nausea POD 2	16 (45.6)	5 (27.8)	0.987
Nausea POD 3	12 (22.2)	6 (33.3)	0.049
Mobilisation POD 0	33 (61.1)	9 (50.0)	0.554
Mobilisation POD 1	31 (57.4)	7 (38.9)	0.173
Mobilisation POD 2	40 (74.1)	10 (55.6)	0.153
Mobilisation POD 3	45 (83.3)	11 (61.1)	0.050
Complication < 30 days	38 (70.4)	12 (66.7)	0.768
Complication > 30 and <90 days	24 (45.3)	5 (27.8)	0.192
Median time to passage of stool in days (IQR)	3.0 (2.0–4.0)	2.0 (1.0–4.0)	0.411
Median time to tolerating solid food in days (IQR)	4.5 (2.0–6.3)	5.0 (2.8–7.3)	0.987
Median time to recover ADL in days (IQR)	6.0 (4.0–8.0)	6.5 (4.8–9.0)	0.976
Median time to termination of urinary drainage in days (IQR)	5.0 (2.0–10.5)	8.5 (3.8–13.0)	0.798
Median time to functional recovery in days (IQR)	7.0 (5.0–14.0)	10 (7.0–14.3)	0.328
Median length of postoperative ICU stay in days (IQR)	1.0 (0.0–1.0)	2.0 (1.0–2.3)	0.004
Median length of hospital stay in days (IQR)	7.0 (5.5–14.0)	10.0 (6.8–15.5)	0.440

* epidural included, iv or oral opioids.

**Table 4 cancers-15-04523-t004:** Functional recovery and complications.

Functional Outcomes and Complications	*n* = 72 (%)
Median time to oral pain control in days (IQR)	4.0 (3.0–4.0)
Median time to passage of stool in days (IQR)	3.0 (2.0–4.0)
Median time to tolerating solid food in days (IQR)	5.0 (2.0–7.0)
Median time to recover ADL in days (IQR)	6.0 (4.0–8.0)
Median time to termination of urinary drainage in days (IQR)	6.0 (3.0–12.0)
Median time to termination of resection-site drain in days (IQR)	3.0 (2.0–4.0)
Median time to termination of nasogastric tube in days (IQR)	3.0 (2.0–6.5)
Mobilisation according to protocol	47 (65.3)
Median length of hospital stay in days (IQR)	9.0 (6.0–14.0)
Complications	51 (70.8)
Complications < 30 days	44 (61.1)
Most severe complication < 30 days (Clavien–Dindo)	
None	21 (29.2)
I–IIIa	39 (54.2)
IIIb–IV	12 (16.7)
Complications > 30 days <90 days	29 (40.3)
Gastro-intestinal complications	
Gastroparesis/paralytic ileus	18 (25.0)
Intra-abdominal abscess	8 (11.1)
Leakage anastomosis	6 (8.3)
Wound infection/dehiscence	12 (16.7)
Mechanical ileus	4 (5.6)
Urological complications	
Bladder retention	5 (6.9)
Urinary tract infection	9 (12.5)
Other	6 (8.3)
Neurological complications	9 (12.5)
Cardio-pulmonary complications	10 (13.9)
Vascular complications	3 (4.2)
Reoperations	14 (19.4)
Readmissions	19 (26.4)

## Data Availability

Data is available upon request to the corresponding author with specification of the purpose of the request.

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
