# Peer review of "Implementation of an Enhanced Recovery after Surgery Protocol in Advanced and Recurrent Rectal Cancer Patients after beyond Total Mesorectal Excision Surgery: A Feasibility Study"

_cancers, 2023, doi:10.3390/cancers15184523_

Round 1

Reviewer 1 Report

Thank you for the opportunity to review this interesting study. My comments are as follows:

1.       The study methodology is well described and the manuscript is written in a logical and eloquent fashion.

2.        The topic is important and of current relevance.

3.       The tables are extremely detailed and comprehensive.

4.       Over the study period, were there any patients who underwent surgery for LARC or LRRC who were not on the ERAS pathway? Or was ERAS automatically implemented for all patients with LARC and LRRC?

5.       Were there any difficulties convincing attending surgeons to allow patients to placed on ERAS protocols? Many surgeons have traditional mindsets and may not be comfortable especially for patients with complex disease. How was this problem overcome?

6.       It is often difficult to coordinate all aspects of a multimodal ERAS programme. Was there a dedicated ERAS nurse or clinical champion to help coordinate ERAS components and collect data?

7.       On the whole, this is an excellent study of the feasibility of ERAS in a challenging group of patients. I congratulate the authors.

Author Response

First and foremost, thank you for your review of our work and engaging comments. In the period from October 2021 until December 2022, all patients with LARC and LRRC who underwent bTME surgery, were included in the study without exceptions. We used a period of 3 months to implement and educate all involved caretakers intensively, to assure the best possible compliance straight from starting the ERAS protocol.

We had a succesful implementation of ERAS in colorectal surgery in the past years. We used the experience of this implementation process for the next step; ERAS in bTME surgery. In the beginning, we indeed experiences some minor difficulties with implementation. As current literature on the use of ERAS in bTME surgery is scarce, we had minimal leverage to persuade everyone. We therefore analysed the data frequently and presented the outcomes in the dedicated multidisciplinary team. Especially the multimodal analgesia protocol was most difficult to implement, therefor frequently reporting and discussing the data was essential to make the implementation a success.

As the results of the protocol are promising, we think the effects on functional outcome and complication rates will only increase when compliance is even greater over time.

The surgeons of our centre already knew the results from the colorectal ERAS and were reluctant to see some form of standardisation in these complex patients as well. The coordination of the program was challenging, but we had a dedicated member of every staff group (“ERAS dedicators”: surgeon, anaesthesiologist, ICU physician, intensive care nurse, ward nurse, nurse practitioner, stoma care nurse, dietician, physiotherapist, and surgical resident). We analysed the results frequently, and presented this data with the improvements to the ERAS dedicators, who passed this on to their team.

We added to the manuscript: “As with all new protocol implementations, compliance will increase over time by gaining more affinity and experience with the protocol. In our centre, care takers were already familiar with ERAS, which facilitated the implementation. Even so the compliance was already >70%, it should increase even further over time, by the increased affinity of the care takers with it, as well as the continuous evolution of the protocol.” (line 260-265)

Thank you again for your review of our manuscript, we truly value your point of view and could not agree more with the difficulties we face when implementing a new standard of care.

Reviewer 2 Report

It is great to read about a particular cohort of patients, hard to standardise for variation in type of surgery. This is the biggest plus for this manuscript, extending standard protocol to where one would expect only an individual approach could work. Even in this specific but relatively heterogeneous group, results suggest that the modified but standardized ERAS approach might benefit patients more. 

That said, presentation is clear apart from the slightly overwhelming tables, and might profit from a more succinct version of them. 

Author Response

Thank you very much for reviewing our manuscript. We could not agree with you more, that implementing a new protocol in these patients is rather difficult. However, we found that even in these complex patients, standardisation of care is possible, and even improving functional recovery and postoperative complications.

The tables are indeed somewhat overwhelming, so we slightly reduced them to present the most important details.